# Antibacterial, Antifungal, Antiviral, and Antiparasitic Activities of *Peganum harmala* and Its Ingredients: A Review

**DOI:** 10.3390/molecules27134161

**Published:** 2022-06-29

**Authors:** Zihao Zhu, Shujuan Zhao, Changhong Wang

**Affiliations:** The MOE Key Laboratory for Standardization of Chinese Medicines, Shanghai Key Laboratory of Compound Chinese Medicines, Institute of Chinese Materia Medica, Shanghai University of Traditional Chinese Medicine, Shanghai 201203, China; zh6876@126.com (Z.Z.); zhaoshujuan@shutcm.edu.cn (S.Z.)

**Keywords:** antimicrobial, antibacterial, antifungal, antiviral, genus *Peganum*, *Peganum harmala*, review

## Abstract

Infectious diseases have always been the number one enemy threatening health and well-being. With increasing numbers of infectious diseases, growing resistance of pathogens, and declining roles of antibiotics in the treatment of infectious diseases, it is becoming increasingly difficult to treat new infectious diseases, and there is an urgent need to develop new antibiotics to change the situation. Natural products tend to exhibit many special biological properties. The genus *Peganum* (Zygophyllaceae) has been used, for a long time, to treat cough, asthma, lumbago, hypertension, diabetes, and Alzheimer’s disease. Over the past two decades, a growing number of studies have shown that components from *Peganum harmala* Linn and its derivatives can inhibit a variety of microorganisms by inducing the accumulation of ROS in microorganisms, damaging cell membranes, thickening cell walls, disturbing cytoplasm, and interfering with DNA synthesis. In this paper, we provide a review on the antibacterial, antifungal, antiviral, and antiparasitic activities of *P. harmala*, with a view to contribute to research on utilizing *P. harmala* for medicinal applicaitons and to provide a reference in the field of antimicrobial and a basis for the development of natural antimicrobial agents for the treatment of infectious diseases.

## 1. Introduction

Infectious diseases pose significant harm to the health of humans, animals, and plants, and they occur all over the world because of the diversity and strong infectivity of their pathogens. The effective prevention and treatment of infectious diseases are major medical problems. For example, the spread of COVID-19 to more than 200 countries and regions around the world in 2020 caused profound influences on human healthy life and survival [1,2] and, as of 28 July 2021, the World Health Organization (WHO) data showed that the number of confirmed cases of COVID-19 in the world reached 195 million, with a cumulative death toll of 4.18 million, representing an infectious disease with the largest death toll since SARS in 2002 and the greatest impact on human health [3,4].

Antimicrobial drugs play an important role in the prevention and treatment of infectious diseases [1,2]. According to statistics, in the past seven decades since the discovery, mass production, and clinical use of antibiotics, billions of patients have been saved worldwide [5]. Despite continuous developments and progress in medical science, many pathogens have evolved drug resistance to antimicrobial drugs due to genetic changes, reducing the efficacy of antibiotics and significantly limiting clinical applications. In addition, the development of antibiotics has almost stagnated in the past decade due to long R&D cycles of antibiotics, high R&D costs, and low commercial returns [6,7]. Therefore, there is an urgent need for new antimicrobial drugs to address diseases caused by drug-resistant microorganisms.

In recent years, numerous studies have shown that more and more natural products of plants have special biological activities as natural antibiotics that play important roles in prevention, treatment, and reducing disease prevalence [8,9,10,11], such as the artemisinin extracted from herb *Artemisia annua* L. with antimalarial effect [12].

*Peganum harmala* L., a perennial herb that belongs to the Zygophyllaceae family, is distributed in the Mediterranean region of Europe, Central Asia, and southern South America. It is commonly used as winter feed for cattle, sheep, camels, and other livestock and also as a traditional Chinese medicine for treating a variety of human diseases [13]. Phytochemical studies have discovered that the main chemical components in the plant are alkaloids, flavonoids, volatile oils, and trace elements [14]. *P. harmala* has many clinical pharmacological effects, such as antibacterial, anti-inflammatory, antibacterial, leukemia resisting, psoriasis resisting, and memory enhancement, and has been clinically used for treating cough, hypertension, diabetes, jaundice, colic, lumbago, malaria, Alzheimer’s disease, and other human diseases [13]. Reports on the antibacterial, antifungal, antiviral, and antiparasitic activities of *P. harmala* have been increasing each year. However, to the best of our knowledge, the antimicrobial effects and mechanisms of *P. harmala* have not been reported in detail, and therefore, it has been difficult for researchers to understand the antimicrobial effects of *P. harmala* and to develop its R&D and application potential in the antimicrobial field. In this paper, we review the antibacterial, antifungal, antiviral, and antiparasitic activities and mechanisms of *P. harmala* extracts and its ingredients by retrieving the relevant literature in globally recognized databases such as Web of Science, PubMed, Google Scholar, Elsevier, and Chinese National Knowledge Infrastructure (CNKI). This literature review should provide a reference for research on the utilization of *P. harmala* for medicinal purposes and the basis for the development of natural antimicrobial drugs.

## 2. Biological Characteristics of *P*. *harmala*

*P. harmala*, is also known as “smelly ancient flower” in the Xinjiang Uyghur Autonomous Region, China; it is a glabrous perennial herb plant that is 30–70 cm high. It has numerous roots that are up to 2 cm thick and stems that are erect or spreading with numerous branched from the base. Its leaves are alternate, ovate, and divided into 3–5 lanceolate-striate lobes that are 1–3.5 cm long and 1.5–3 mm wide [15]. The morphology of *P. harmala* is shown in Figure 1.

## 3. Main Components of *P. harmala*

As early as 1900, a researcher studied the inhibitory effects of total alkaloids from *P. harmala* on retinoblastoma cells. People have studied the pharmacological, chemical, and biological activities of *P. harmala* for more than 100 years. There are many different compounds in *P. harmala* [14,16], which were reported in detail in 2017 [13]. More than 308 compounds have been isolated from *P. harmala*, including 97 alkaloids, 24 flavonoids, 10 triterpenoids, 3 anthraquinones, 2 phenylpropanoids, 18 carbohydrates, 17 amino acids, 99 volatile oils, 26 fatty acids, 3 sterols, 1 vitamin, 1 protein, 1 carotene, and 6 other trace elements. Among these compounds, the highest content is β-carboline alkaloids (βCs). The alkaloid content is up to 10% in seeds, followed by roots, and the least amount in leaves. They main compounds include harmine, harmaline, harmalol, harmane, and harmol; a few new compounds have been reported in recent years.

In 2017, Wang et al. [17] conducted a study on the chemical composition of *P.*
*harmala* seeds. The seeds of *P.*
*harmala* were extracted by ethanol reflux, and then the extract was concentrated, the pH adjusted, and separated by silica gel column to obtain nine compounds. Among them, (−)-peharmaline A (**1**) (Figure 2) and (+)-peharmaline A (**2**) (Figure 2) were rare carboline-vasicinone hybrid alkaloid enantiomers with an unknown hybrid dimer system.

Wang et al. [18] extracted the seeds of *P. harmala* with ethanol reflux, adjusted the pH of the extract, extracted it, and passed it through the silica gel column. Six new βCs were identified, i.e., pegaharmines F–K (**3**–**8**) (Figure 2).

In a study by Li et al. [19], two pairs of new alkaloid glycoside dimorphisms (**9**–**12**) (Figure 2) and a new enantiomer (**13**) (Figure 2) were identified in the ethanol extracts of *P. harmala* seeds. The names of the compounds **9**–**12** are (*S*)- and (*R*)-1-(2-aminobenzyl)-3-hydroxypyrrolidin-2-one *β*-D-glucopyranosyl-(1→6)-*β*-D-glucopyranoside (**9**,**10**), (*S*)- and (*R*)-vasicinone *β*-D-glucopyranosyl-(1 → 6)-*β*-D-glucopyranoside (**11**,**12**).

In 2019, Fang et al. [20] extracted seeds of *P. harmala* with 95% ethanol and concentrated them. The concentrate was re-extracted with n-hexane and ethyl acetate. The extracts were subjected to silica gel column chromatography, ODS column chromatography, MCI resin, and other technologies, and a new compound was found and named N-[3-(2-amino-4-methoxyphenyl)-3-oxopropyl] acetamide (**14**) (Figure 2).

In 2020, Wu et al. [21] isolated six new compounds against HSV-2 virus from crude alkaloids of *P. harmala* seeds and identified the structures of six new βCs based on HR-ESI-MS data, named pegaharines A–F (**15**–**20**) (Figure 2).

## 4. Antimicrobial Activity

### 4.1. Antibacterial Activity

The antibacterial effects of *P. harmala* have been demonstrated in many Gram-positive and Gram-negative bacteria.

Ahmad et al. [22] determined the antibacterial activity of harmine, harmaline, and their derivatives using the broth incorporation method with tryptone soya broth (Oxoid) and recorded the minimum inhibitory concentrations (MICs). They found that the MICs of harmine against *Bacillus pumilus*, *B. subtilis*, *Corynebacterium hofmannii*, *Sarcina lutea*, *Staphylococcus citreus*, *S. lactis*, *Aeromonas hydrophila*, and *Salmonella paratyphi* A were 100 μg/mL, respectively. This was the same activity as harmaline against *C. hofmannii* and *S. lutea*.

Abutbul et al. [23] studied the antibacterial (*A. hydrophila*, *Streptococcus iniae*, and *Vibrio alginolyticus*) activity of water extracts from desert plant extracts (including *P. harmala*) on fish pathogenic bacteria in vitro using the disk diffusion method. It was found that *P. harmala* seeds extract had the highest inhibition effects on *A. hydrophila* and *V. alginolyticus*; the inhibition area ranged from 18.0 to 20.5 mm.

Muhaisen et al. [24] observed the in vitro antibacterial *(Bacillus cereus* ATCC 11778, *B. subtilis* ATCC 6633, *Enterococcus faecalis* ATCC 29212, *E. coli* ATCC 25922, *P. aeruginosa* ATCC 27853, *S. aureus* ATCC 25923, and *S. epidermidis* ATCC 12228) activities of ethanol, hexane, chloroform, and methanol extracts from eight medicinal plants (including *P. harmala* leaves) using broth microdilution. It was found that there was inhibitory activity of *P. harmala* against Gram-negative bacteria (MIC range of 4–8 mg/mL) and the chloroform extract from *P. harmala* leaves. The results showed the strongest inhibition of extract from *P. harmala* leaves against *P. aeruginosa* with MICs ranging from 0.25 to 1.0 mg/mL.

Arshad et al. [25] tested the activities of methanol, ethanol, and water extracts of 12 medicinal plants (including *P. harmala*) against 20 bacteria (*Acinetobacter* sp. 7987, *Clostridium* sp. 1729, *E. coli*-SN-11, *E coli*-SN-07, *E. coli*-5457, *E. coli*-6058, *E. coli*-5706, *E. coli*-1253, *E. coli*-1303, *E. coli*-3931, *Pasteurella multocida* 1294, *Staphylococci* sp. 772, *Streptococci* sp. 1959, *Proteus* sp. 6433, *Salmonella* sp. 2853, *Salmonella* sp. 3102, *Salmonella* sp. 4922, *Salmonella* sp. 4377, and *Salmonella* sp. 3402) by performing disk and agar diffusion experiments. The extract of *P. harmala* seeds was found to inhibit the growth of all bacteria at concentrations of 0.38~1.65 mg/mL. The antimicrobial activities of crude extracts and ingredients from *P. harmala* were further compared using the microdilution method and the order of antimicrobial activities was harmane, harmaline, harmalol, and harmine. It was suggested that *P. harmala* or the ingredients could play a significant role in drug development for controlling bacteria.

Darabpour et al. [26] examined the antibacterial activity of methanol extracts from different parts of *P. harmala* (root, stem, leaf, flower, and seed) against 11 bacteria (Gram-positive bacterial species of *Bacillus anthracis*, *B. cereus*, *B. pumilus*, *S. aureus*, *S. epidermidis*, *Listeria monocytogenes,* and *S. pyogenes*, and Gram-negative bacterial species of *P. aeruginosa*, *Brucella melitensis*, *P. mirabilis*, *S. typhi*, *E. coli,* and *K. pneumoniae*) using disk diffusion. It was found that root extract had better antibacterial activity against Gram-positive bacteria than seed extract. The antibacterial activity of the leaf was medium, while those of the stem and flower were weak. The MIC and MBC of the root and seed extracts against MRSA were 0.625 mg/mL, which was similar to the seed extract against *E. coli* and *S. typhi*. It was also found that seed and root extracts showed synergistic effects when combined with neomycin, colistin, and carbenicillin. In conclusion, harmine could be used in the development of drugs for the treatment of infectious diseases caused by MRSA.

Omar et al. [27] investigated the anti-*Cronobacter sakazakii* (ATCC 29004) activity of methanol and petroleum ether extracts from six plants (including *P. harmala*). It was found that the methanol extract of *P. harmala* effectively inhibited the growth of the strain, with an MIC of 0.00375 mg/mL. The presence of 2% reconstituted infant milk formula could reduce the inhibitory effect of plants on the growth of *C. sakazakii*, and the mixture of nisin and disodium ethylenediamine tetraacetate had a slight synergistic effect on the inhibition of *C. sakazakii*.

Fazal et al. [28] studied the antibacterial (*S. aureus* 6538, *P. aeruginosa* 9721, *E. coli* 25922, *K. pneumoniae*, *Salmonella typhi*, *B. subtilis*, and *B. cereus*) activities of 11 medicinal plants including *P. harmala,* using the disk diffusion method. It was found that the ethanol and hexane extracts of *P. harmala* showed the optimal inhibitory activity against *K. pneumoniae* (the inhibition zone was 25.8 mm), while the inhibition zone against *E. coli* was 10.8 mm. The chloroform extract was the most effective against *S. aureus* (inhibition zone of 24.5 mm). It was indicated that plants, such as *P. harmala*, could be used in the development of antibiotics for the treatment of diseases caused by *K. pneumoniae*.

Soltani et al. [29] studied the activity of essential oil and aqueous extracts of 24 medicinal plants (including *P. harmala*) against *Xanthomonas arboretum* pv *juglandis*. Most of the plant extracts were observed to possess antibacterial activity using an agar disk diffusion test. The aqueous extracts of six plants had significant antibacterial activity with antibacterial area of 6.0 mm. It was concluded that the aqueous extracts of these plants had broad possibilities in plant disease control and could be used as natural biological control agents in walnut orchards.

Irshaid et al. [30] identified the activities of methanol extracts from aerial parts of four medicinal plants (including *P. harmala*) against five bacteria (*S. aureus*, *E. coli*, *P. aeruginosa*, *E. cloacae*, *P. mirabilis*) using broth dilution and a disk diffusion test. The results showed that *P. harmala* extracts inhibited the growth of these bacteria with MICs of 0.8, 1.2, 0.9, 1.0, and 0.9 mg/mL, respectively.

Apostolico et al. [31] performed an agar diffusion test to evaluate the antibacterial activities of essential oils from *P. harmala* seeds from Algeria, Egypt, Libya, Morocco, and Tunisia, on *Bacillus cereus* 4313, *B. cereus* 4384, *E. coli* 857, *Pseudomonas aeruginosa* 50071, and *S. aureus* 25693. The results showed that all samples inhibited the growth of the bacteria. *E. coli* showed the highest sensitivity to these oils, especially the oil of *P. harmala* from Egypt at a concentration of 15 μg/mL with an inhibition zone of 10.0 mm, which was higher than that of the control group (tetracycline, inhibition zone of 10.0 mm). It was speculated that this may be due to the differences in the growth environment of *P. harmala* in different countries, resulting in different percentages of secondary metabolites of plants such as oxygen-containing monoterpenes, sesquiterpenes, and oxygen-containing sesquiterpenes. These findings were considered to be important in an era of serious problems with antimicrobial resistance.

The ethanol extract of *P. harmala* seeds showed an inhibitory effect (MIC of 4 μg/mL) on *E. faecalis*, but the inhibitory effect was not significantly different from that of 0.5% NaOH [32]. It is expected to become a safe therapeutic agent. Khalid et al. [33] assessed the anti-*S. aureus* and anti-*P. aeruginosa* activities of low-molecular-weight peptides in seeds and leaves of 20 plant species (including *P. harmala*) using a dish diffusion assay. *P. harmala* peptides (PhAMP) isolated from *P. harmala* had a maximum zone of inhibition against the two laboratory bacterial strains. Then, the authors studied the antibacterial potential of PhAMP against pathogens in burn wounds (*S. aureus*, *P. aeruginosa,* and *K. pneumoniae*) and surgical wounds (*P. aeruginosa* and *K. pneumoniae*). It was found that PhAMP was effective at disrupting biofilm formation of all pathogens after 36 h of treatment. These data indicate that *P. harmala* has the potential to be developed into a natural antibiotic drug.

Ait Abderrahimet al. [34] measured the antibacterial activities of the methanol extracts from *P. harmala* and *Ziziphi spinosae* against four pathogenic microorganisms (*S. aureus*, *E. coli*, *Candida albicans,* and *P. aeruginosa*) and used an agar dilution method to determine the MICs. It was found that the extracts of *P. harmala* seeds showed growth inhibition against all the test strains, with MIC values of 0.5, 1.0, and 6.0 mg/mL, respectively. The antibacterial effect of methanol extracts from *P. harmala* was better than that of *Z. jujuba*. It was demonstrated that *P. harmala* was an effective antibacterial agent. In the same year, Iranshahy et al. [35] assessed the antibacterial activity of the chloroform extracts of the fruits and flowers of *P. harmala* against five microorganisms including *M. luteus* by using the disk diffusion method. They found that the total alkaloids had a strong specificity for *M. luteus* and a low sensitivity to Gram-negative bacteria, especially *P. aeruginosa*.

Nenaah et al. [36] evaluated the antibacterial activities of four βCs (harmane, harmine, harmaline, and harmalol) from *P. harmala* seeds using the disc diffusion method with high potency biodisc. In separate alkaloid resistance tests, *Escherichia coli*, *P. vulgaris*, *S. aureus,* and *Bacillus subitilis* were proven to be the most susceptible to inhibition by harmane, harmine, harmaline, and harmalol, with inhibition zones of 17.9, 24.7, 14.7, and 21.1 mm and MICs of 0.50, 0.83, 1.00, and 0.75 mg/mL, respectively. Harmane had the highest activity against *E. coli* and harmalol had moderate antibacterial activity. When used with the binary mixture of harmane and harmaline, the inhibitor was the most potent against *P. vulgaris* (inhibitory zone of 28.9 mm and MIC of 0.41 mg/mL) and against *B. subitilis* (inhibitory zone of 26.1 mm and MIC of 0.33 mg/mL). In addition, it was concluded that the selection of compounds that act synergistically with *P. harmala* alkaloids had great potential in the treatment of diseases caused by *P. vulgaris*. Shaheen et al. [37] evaluated the inhibitory activity of *P. harmala* seed alkaloid extracts against four plant pathogens (*R. solanacearum Physiotype* II, *Erwinia Amylovora*, *Pectobacterium Carotovorum* subsp., and *Burkholderia gladioli*) in vitro using the agar diffusion method. *R. solanacearum phylotype* II was the most sensitive to the extract (MBC of 150 μg/mL), followed by *B. gladioli* (MBC of 200 μg/mL). When the concentration was reached between 4 and 300 μg/mL, the extract showed a significant inhibitory effect against *R. solanacearum*. Transmission electron microscopy revealed that bacterial wilt cells were severely damaged, with coagulated genome, thickened cytoplasm, cell wall, and disorganized structure. In vivo studies indicated that extracts of 300 μg/mL reduced plant brown spot symptoms. It was concluded that total alkaloid extract of *P. harmala* could be an alternative to chemical antimicrobial agents in the treatment of these bacterial diseases. Siddique et al. [38] also tested the in vitro inhibitory effect of seven plant extracts including *P. harmala* on bacteria using a paper disc diffusion method and in vivo inhibitory effect using pot experiments. Oxytetracycline treatment was a positive control. The extracts (100%, 75%, and 50% concentrations) of *P. harmala*, *M. piperita*, *A. sativum*, *W. somnifera*, *M. azedarach*, *C. processra,* and *N. oleander* could inhibit the growth of *C. michiganensis*. The undiluted water extract of *P. harmala* had the highest antibacterial activity in these plants, and the inhibition zone was 14.40 mm. The positive control oxytetracycline (200 ppm) showed the highest antibacterial activity inhibition zone, which was 24.70 mm. In vivo experiments showed that after 56 days of treatment, the plant disease degree of *P. harmala* dry powder treatment was significantly reduced, and the plant height was 10.33% higher than that of the plant treated with clarithromycin. It was suggested that dry powder or extract from these plants could be used to control canker of tomato.

The inhibitory activity of the n-butanol extract of *P. harmala* seeds against three species of *P. aeruginosa* was superior to that of cefazolin and vaamox [39]. When incubated with three *P. aeruginosa* at 500 μg/mL for 4 h, the sterilization rate reached 100%, and the compound further isolated is harmaline.

*P. harmala* seeds smoke can act as an air disinfectant to reduce the concentration of bacteria in air [40]. The removal rate of bacteria in air after 5 g of seeds in residential areas produced smoke for 5 min reached a maximum of 71.4%. In an educational setting, the bacterial removal rate in the air after 10 min of smoke from 10 g of seeds reached 92.8%.

Therefore, so far, there is an inhibitory effect of *P. harmala* extract on a variety of pathogenic bacteria that cause disease in animals and plants, with a wide antibacterial spectrum. The potential of its resistance to plant pathogens needs further exploration. The extracts of seeds and roots have good antibacterial effect as compared with some antimicrobials, which provides clues for future studies on synergistic antibacterial with other drugs.

### 4.2. Antifungal Activity

The antifungal effects of *P. harmala* have been demonstrated in various pathogenic fungi.

Ahmad et al. [22] tested 16 fungi with harmine, harmaline, and their derivatives at a concentration of 50–500 μg/mL on Sabouraud dextrose agar (Oxoid) slants, and recorded inhibition zones to compare with the MICs. It was found that harmine of 100 μg/mL was effective against all eight dermatophytes used (*Epidermophyton floccosum*, *Microsporum canis*, *T. longjifisis*, *T. mentagrophytes*, *T. rubrum*, *T. simii*, *T. tonsurans*, and *T. violaceum*), while harmaline inhibited *M. canis*, *T. longjifisis*, *T. rubrum*, *and T. tonsurans* at 500 μg/mL. Other compounds reduced the growth of some fungi at higher concentrations as compared with the control group (terbinafine of 0.01–25 μg/mL).

Shahverdi et al. [41] mixed different amounts (0.156, 0.312, 0.612, 1.25, 2.5, and 5 mg) of dichloromethane and n-hexane extracts of smoke from *P. harmala* smoldering seeds with Mueller–Hinton agar, and determined the inhibition zone of fungi (*Aspergillus niger* PIM, *C. albicans* ATCC 14053, and *Cryptococcus neoformans* kf 33) using a conventional disk-diffusion method. It was found that there was no antibacterial activity of the dichloromethane extract at an amount of 0.156 mg, but higher contents showed good antifungal activities against *S. epidermidis* and *C. neoformans*.

Sarpeleh et al. [42] studied the inhibition of aqueous, alcoholic, and methanolic extracts of leaves, floral tissues, and seeds of *P. harmala* to 13 fungal (*Alternaria* sp., *Botrytis cinera*, *Cladosporium cucumerinum*, *Corynespora cassiicola*, *Fusarium oxysporum* f.sp melonis, *Macrophomina phaseolina*, *Monosporascus annovballus*, *Phytophthora drechsleri*, *Rhizoctonia solani*, *Sclerotinia sclerotiorum*, *Trichoderma harzianum*, *Ulocladium* sp., and *Verticillium dahliae*) species using mycelial growth inhibition assay. It was found that the mycelium growth rate of most fungi decreased when treated with water extract and methanol extract, and the seed extract had the highest activity. The water-soluble seed extract could inhibit the spore germination of *F. oxysporum*. The authors suggested that extracts of *P. harmala* could be used as a substitute for antifungal agents. Another study also illustrated this point [43].

Nenaah et al. [36] determined the activities of four alkaloids (harman, harmine, harmaline, and harmalol) from *P. harmala* seeds against two fungi (*A. niger* and *C. albicans*) using the disk diffusion method. It was found that harmaline had the best inhibitory effect on *C. albicans*, with inhibitory zones between 21.2 and 24.7 mm. When used in the form of total alkaloid mixture, the inhibition zone reached 31.5 mm. Its binary mixture was recommended for use as a novel antifungal agent.

Diba et al. [44] studied the activities of alcohol extract of *P. harmala* seeds against *C. glabrata* and *C. tropicalis* using serial dilutions in tubes and serial dilutions in agar media. It was confirmed that the MIC of *P. harmala* extract against *C. glabrata* was 0.312 mg/mL and the MFC was 0.62 mg/mL. The MFC against *C. tropicalis* was 0.125 mg/mL. It was suggested that there was antifungal activity of *P. harmala* seed extract against opportunistic yeast and *Candida* spp.

The conidial suspension of two fungi (*Penicillium digitatum* and *B. cinerea*) was prepared in buffers of pH 5 and pH 9, and treated with 1 mM harmol for 24 h to evaluate the number of pathogens. As compared with the control, the counts of pathogens decreased by two times at pH 5, and the spore viability was completely lost [45].

Hajji et al. [46] mixed oil of *P. harmala* seeds at concentrations of 50%, 25%, 12.5%, 6.25%, 3.125%, 1.562%, 0.781%, 0.39%, and 0.195% with melted agar and measured the hyphae to study the inhibitory effect of different concentrations of oil to 10 fungi (*R. solani*, *M. phaseolina*, *Pythium* sp*. 1*, *Pythium* sp*. 2*, *Alternaria* sp., *Colletotrichum* sp., *M. cannonballus*, *F. solani* f. sp*. cucurbitae*, *F. oxysporum* f. sp*. melonis*, and *F. oxysporum* f. sp. *niveum*). The results of mycelial measurements revealed that the oil of *P. harmala* seeds at a concentration of 50% had good activity against *Pythium* sp. with inhibition rates of mycelial growth ranging from 56% to 82%, followed by *F. solani* f. sp. *cucurbitae* with inhibition rates ranging from 15% to 55.7%. Seed oil had a moderate inhibitory effect on *Colletotrichum* sp. with the inhibition from 5.19% to 42%. It was concluded that *P. harmala* was expected to be a future drug for controlling some microorganisms.

Izadi et al. [47] extracted the dry leaves and fruits of *P. harmala* with 70% ethanol. After removing organic solvents, chitosan nanoparticles were used to encapsulate the extracts and forsythia essential oil (NCE), and the activity of encapsulated nanoparticles against *Alternaria* spp. was evaluated in vitro and in vivo. The results showed that the antifungal activity of *P. harmala* extract at 100 ppm was similar to that at 1000 ppm, and the fungus was completely inhibited at 750 ppm. Encapsulated with forsythia essential oil, the total inhibitory concentration decreased to 200 ppm. It was speculated that chitosan attached to the cell membrane near the release of content and enhanced antifungal activity. *P. harmala* seed smoke reduced the number of fungi in the air. After 30 min of exposure to air from the smoke generated by 10 g seeds, the fungal removal rate reached 94.7% [40].

From the current studies, it can be seen that there are many antifungal studies on *P. harmala*, which have revealed that there is good antifungal activity of *P. harmala*. The active part is mostly the crude extract of seeds. However, studies on screening antifungal compounds and structure optimization of high antibacterial compounds are rare. It is an urgent need to determine the optimization of active compounds and lead compounds in future studies.

### 4.3. Antiviral Activity

The inhibitory effects of *P. harmala* on a variety of animal viruses and plant viruses have been reported.

Ma et al. [48] prepared *P. harmala* seed crude protein extract (PHP) with 50–80% ammonium sulfate and tested the anti-HIV-1 RT activity of PHP according to the instructions of the HIV-1 RT kit, which showed a maximum inhibition rate of 69.1% at a PHP concentration of 3.75 μM and estimated an IC_50_ value of 1.26 μM.

Song et al. [49] tested the anti-TMV activities of β-carboline, dihydro-β-carboline, and tetrahydro-β-carboline alkaloids and their derivatives using the Ishida’s method. It was found that all alkaloids and some of their derivatives exhibited higher anti-TMV activities than ribavirin in vitro and in vivo. In particular, the in vitro and in vivo activities of harmalan (60.3%) and tetradydroarmane (59.5%) at 500 μg/mL were much higher than that of ribavirin (38.5%).

Moradi et al. [50,51] evaluated the effects of ethanol extract from *P. harmala* seeds and its total alkaloids on MDCK cells infected with influenza A virus (A/Puerto Rico/8/34 (H1N1, PR8) virus), and the mechanism of RNA polymerase blocking was studied. The results showed that there were good inhibition effects of the crude extract and total alkaloids on the virus with the CC_50_ values of 122.9 and 133.9 µg/mL, respectively. The extract reduced viral mRNA expression and inhibited viral protein synthesis in a dose-dependent manner, with no effect on hemagglutination inhibition and virucidal activity.

Wu et al. [21] compared the anti-HSV-2 activities of pegaharine A–F from *P. harmala* seeds with two representative β-carboline alkaloids (harmine and harmaline). Pegaharine D showed the strongest antibacterial activity of these compounds with an IC_50_ value of 2.12 ± 0.14 μM.

Edziri et al. [52] evaluated the antiviral activities of extracts of *P. harmala* leaves against the HCMV strain AD-169 and Coxsackie B virus type 3 (CoxB-3). Except for the petroleum ether extract of *P. harmala* leaves without resistance, there was anti-HCMV activities of all the other extracts. The methanol extract had the strongest activity. Antiviral activities varied from 80% to 95% at concentrations from 25 to 100 mg/mL. In the same year, Chen et al. [53] reported that harmine from 502 natural products was the most effective agent for EV71 virus treatment in vitro with CC_50_ values between 400.0 and 500.0 μM. Harmine at concentrations of 10, 30, and 100 μM was observed under an inverted microscope to inhibit viral activity, and to downregulate the RNA and protein levels of EV71. It reduced the cytopathogenic effect in a dose-dependent manner. DCFH-DA probe measurements showed that harmine inhibited ROS production in a dose-dependent manner, which was related to the downregulation of NF-κβ activation.

A complex of harmala has been shown to have highly selective anti-H1N1 activity without affecting host cells. The complex of 2-hydroxypropyl-β-cyclodextrin and harmala loaded into PLGA nanoparticles showed highly selective antiviral activity against H1N1 (IC_50_ of 2.7 μg/mL), with low toxicity to host cells [54].

The current studies have reported the antiviral activity of different components of *P. harmala*. In most of these studies, only the antiviral extracts of *P. harmala* have been reported, but research on screening antiviral compounds in the extracts is rare, and there is less research on the antiviral mechanism of *P. harmala*. Therefore, there is still more research needed on these aspects.

### 4.4. Antiparasitic Activity

*P. harmala* has been proved to have significant anti-parasitic activity. Lasa et al. [55] studied the antileishmanial activities of different forms of harmine through a hamster model. It was found that free, liposome, vesicle, and nano-articular forms of harmine reduced the parasitic load of the spleen by about 40%, 60%, 70%, and 80%, respectively. A cell cycle analysis study using flow cytometry showed that harmine interfered with the cell division stage and confocal microscopy showed that cell death was caused by non-specific membrane damage.

Mirzaei et al. [56] treated 50 cattle naturally infected with *Theileria annulata* with extract of the aerial parts of *P. harmala* at a dose of 5 mg/kg for 5 days. After about 15 ± 3 days, the symptoms and parasites of 39 cattles had disappeared. The recovery rate was 78%. Schizonts of *T. annulata* were not observed in lymph node biopsy smears.

Astulla et al. [57] tested the anti-*P. falciparum* activities of four alkaloids (harmine, harmaline, vasicinone, and deoxyvasicinone) from *P. harmala* seed extract and found that there was a moderate inhibition of harmine (IC_50_ value of 8.0 μg/mL) and harmaline (IC_50_ value of 25.1 μg/mL) on *P. falciparumin* in in vitro. Vasicinone and deoxyvasicinone did not display inhibitory effects on the protozoa (IC_50_ > 10 μg/mL). In human monocyte toxicity studies, it was found that harman and tetrahydroharman reduced THP1 cells in the S phase of the cell cycle, presumably inhibiting total protein synthesis [57].

Derakhshanfar et al. [58] infected 5–6-month-old lambs with *T. hirci*, and treated them with the extracts of the aerial parts of *P. harmala* at a dose of 5 mg/kg. Microscopic observation showed that the parasites disappeared in 9 days and the rectal temperature returned to normal in the lymph nodes and peripheral blood smears of lambs in the treatment group. These results proved that there were effects of *P. harmala* extracts in treating sheep tissue damage induced by *T. hirci*.

After 48 h of treatment with 1 mg/mL methanol extract of *P. harmala* seeds, miracidial formation reduced to 0.5%. The methanol extract of *P. harmala* seeds at a dose of 3 mg/mL could inhibit the formation of miracidial [59].

Shoaib et al. [60] incubated *A. castellanii* (ATCC 50492) isolated from a keratitis patient in 24-well plates with three plant extracts, including *P. harmala* seeds, and determined the number and activity of *A. castellanii* using cytometer counting and trypan blue exclusion. The in vitro results showed that *P. harmala* extracts of 1.5 mg/mL had a significant amoebicidal effect on *Acanthamoeba* in a concentration-dependent manner.

Tanweer et al. [61] treated coccidiosis chicken with the methanol extract of *P. harmala* seeds at concentrations of 200 mg/L (PH-200), 250 mg/L (PH-250), and 300 mg/L (PH-300) in drinking water. On the 14th day, chicken were infected with the coccidiosis larvae at a standard dose until the age of 35 days, and the weekly weight gain, feed intake, feed-to-meat ratio, cecal section of the birds were experimentally analyzed. It was found that weight gain, total body weight, and feed-to-meat ratio increased linearly with an increased dose of *P. harmala*. Histopathological observations revealed that the cecal injury and leukocyte infiltration were significantly reduced in broilers of group PH-300. It was indicated that *P. harmala* extract had good anticoccidial effects in broiler chicks.

Shafiq et al. [62] studied the acaricidal activities of methanol extracts of three plants (rhizome of *Curcuma longa*, fruit of *Citrullus colocynthis,* and seed of *P. harmala*) against *R. microplus* using a modified larval immersion method (syringe method). It was found that the activities of all three extracts reached peak values when the dose was 50 mg/mL for 6 days. The lowest acaricidal efficacy was observed at 24 h with *P. harmala* at 3.125 mg/mL alone. Based on these observations, it was suggested that this inexpensive and readily available plant combination formulation could be used on farms.

Tabari et al. [63] collected *Trichomonas gallinae* from pigeons using the wet mount method, cultured it in trypsin/yeast extract/maltose (TYM) medium on multiwell plates with *P. harmala* alkaloid extract, metronidazole, harmine, and harmaline at concentrations of 5, 10, 15, 20, 30, 50, and 100 μg/mL, to determine the MICs. The metronidazole of 50 mg/kg body weight or alkaloids of 25 mg/kg body weight were administered to 60 pigeons experimentally infected with *T. gallinae*. The MICs of *P. harmala* alkaloid extract, metronidazole, harmine, and harmaline were 15, 50, 30, and 100 µg/mL, respectively. Infected pigeons were completely recovered after 3 days of treatment with alkaloids but they were not completely recovered with metronidazole.

At present, *P. harmala* seed extract has shown good inhibitory effects on pathogens and inhibition of some protozoa (*T.* pigeon), which was superior to a positive control drug in some cases. However, there are few studies on its mechanism of action, and the material basis of its efficacy is not clear enough. Therefore, further studies are needed using modern molecular biology and other means.

## 5. Antimicrobial Mechanisms of *P. harmala*

A growing number of studies have shown superior inhibition of *P. harmala* against a wide range of microorganisms, and the antimicrobial mechanisms have also been identified. The antimicrobial mechanisms include: increasing ROS content; destroying the microorganism’s cell membrane and cell wall, causing cytoplasmic disorder; downregulating the expression of bacterial flagellum motility genes and upregulating the expression of capsular polysaccharide synthesis genes and intercellular adhesion genes; inhibiting the generation of microorganisms’ violacein peptide; inhibiting the replication of pathogenic nucleic acids and protein synthesis (Table 1).

### 5.1. Effects on ROS and Cell Membrane

Under normal circumstances, active oxygen in organisms is in dynamic equilibrium, and metabolism imbalance leads to cell membrane damage. The conidial suspension of plant pathogenic fungus *P. digtiatum* was treated with 1 mM harmol. After 24 h, the cell membrane of the conidia was destroyed, the cell membrane was ruptured, and the cytoplasm was disordered. When harmol-treated *P*. *digtiatum* conidia were exposed to UVA, a significant increase in intracellular ROS accumulation was detected by H_2_DCFDA probe [45].

In another experiment using the MTT assay to determine the inhibition of MDR *E*. *coli* by harmine and its derivatives, a number of cell ruptures and cell debris were observed in the treated *E*. *coli*. The lucigenin CL assay showed a dose-dependent increase in ROS in treated *E*. *coli* [64].

PhAMP has good destructive effects on the biofilm formed by pathogenic bacteria. The inhibitory effects of PhAMP on burn and surgical wound pathogens (*S. aureus*, *P. aeruginosa*, *K. pneumoniae*, *P. aeruginosa*, and *K. pneumoniae*) were studied using the disc diffusion method, and then the effect of PhAMP on pathogens isolated from burn-wound biofilm was tested on a 24-well plate. After 36 h of treatment, the formation of crystal violet-stained biofilm decreased and the biofilm was destroyed, as observed under an optical microscope [33]. In another experiment, to evaluate the effect of *P. harmala* extracts on the inhibition of *Acinetobacter baumannii* biofilm, a decrease in biofilm was also observed with scanning electron microscopy [65].

### 5.2. Effects on Nucleic Acid

Nucleic acid is the main information molecule of cells, including deoxyribonucleic acid (DNA) and ribonucleic acid (RNA). In one study, PhAMP was incubated with *P. aeruginosa* and *S. aureus* under shaking for 16 h, and then the expression levels of biofilm-related genes were detected by real-time PCR. It was found that the expression levels of flagellum gene (*flgK*), fimbriae protein gene (*pilA*), and fimbriae gene (*cupA1*) in *P*. *aeruginosa* were significantly downregulated. The expressions of the capsular polysaccharide synthesis gene (*CPS5*) and the intercellular adhesion gene (*icaA*) were upregulated in *S. aureus* [69].

The total alkaloid extract from seeds of *P. harmala* was cultured with inocula of *R. solanacearum* in broth medium. In addition to the thickening of cell wall, disorder of cytoplasm, and serious damage of cells, genome condensation was also observed under transmission electron microscope [37].

In another study, eight plant extracts were mixed with yeast extract sucrose (YES), and the expression levels of aflatoxin B1 synthesis genes were analyzed by RT-PCR. It was found that the reduction in aflatoxin B1 biosynthesis was related to a reduction in (or blocking of) the expressions of *aflR*, *aflM,* and *aflP* by plant extracts such as *P. harmala* [66].

It can also affect virus replication. MDBK cells infected with Bohv-1 were treated with harmine at different stages. It was found that the viral production was significantly reduced, and harmine affected viral replication at early and later stages [67].

### 5.3. Effects on Reverse Transcriptase (RT) Activity

Reverse transcriptase plays a catalytic role in viral reverse transcription. Studies have shown that it can inhibit the proliferation of tumor cells and reduce the activity of HIV-1 RT. A new antifungal protein was isolated from the seeds of *P. harmala* and its inhibition rate on tumor cells was studied according to the cytotoxicity curve. The HIV-1 RT kit was used to test the inhibition effect on HVI-1 RT and it was found that the new antifungal protein could inhibit the proliferation of esophageal cancer, cervical cancer, gastric cancer, and melanoma cells, and the activity of HIV-1 RT was decreased [48].

## 6. Conclusions

*P. harmala* is rich in resources, widely distributed in the world, and its medicinal history is more than 2000 years. Its traditional use in disinfectants and mosquito control has contributed to its successful application in the treatment and prevention of human, animal, and plant diseases. In recent years, epidemics have become the greatest threat to human health and plant quality. The emergence and variation of new pathogens and the emergence of drug resistance pose a huge threat to humans, and therefore, there is an urgent need for effective drugs to change this situation. *P. harmala* extract, harmine, and other βCs have broad-spectrum inhibitory effects on many microorganisms. Therefore, *P. harmala* is a potential source of safe and natural antimicrobial drugs, especially at a time when viruses have become the greatest enemy threatening the safety of human lives.

The antimicrobial activity and antimicrobial mechanism of βCs, such as harmine, were studied to provide a theoretical basis for the development of potential antibacterial drugs. The inhibitory activity of *P. harmala* on broad-spectrum microorganisms may be the reason for its therapeutic effect on many pathogen-related diseases such as wound treatment, skin inflammation, hemorrhoids, and cough. *P. harmala* is a potential resource for the prevention or treatment of plant infectious diseases, such as rust and wilt, due to its insecticidal effect and inhibition of the activity of various plant pathogens.

A rich variety of compounds is the material basis for its broad-spectrum antimicrobial activity. There have been at least 308 chemical constituents isolated and identified in *P. harmala*, including alkaloids, flavonoids, triterpenoids, anthraquinones, phenylpropanes, carbohydrates, amino acids, volatile oils, sterols, vitamins, proteins, carotene, and trace elements, of which alkaloids have better antimicrobial activity tests. However, due to the diversity of plant secondary metabolites and the advancement of technology, new compounds have been discovered from *P. harmala* extracts. The antimicrobial activities of these new compounds need to be further studied. A very meaningful research focus would be to develop efficient broad-spectrum antimicrobial drugs using the compounds in *P. harmala* as antimicrobial scaffolds. In addition, the current research has revealed the biological activities of *P. harmala*, but the application of these biological activities in the treatment of human, animal, and plant diseases is limited, and the application of biological activities in medicine should be further studied in the future.

In conclusion, *P. harmala* inhibits a variety of pathogenic microorganisms and has great potential in the treatment of infectious diseases and air sterilization. Current studies have shown that it contains hundreds of compounds and has a variety of biological activities. These results herald the bright prospect of *P. harmala* as a natural source of next-generation medicines, and also lay the foundation for further elucidation of its therapeutic mechanisms, thus, revealing the relationships among the clinical uses, chemical components, and biological activities of *P. harmala*.

## Figures and Tables

**Figure 1 molecules-27-04161-f001:**
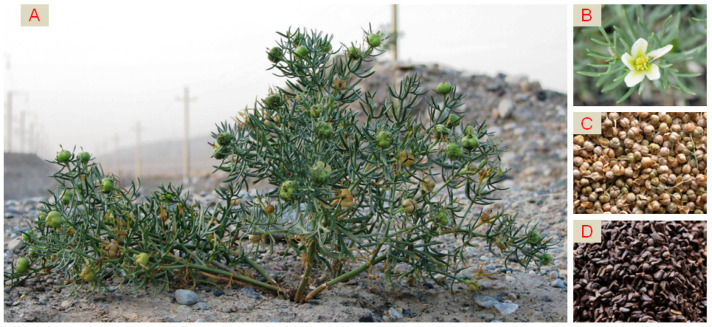
*P. harmala* (**A**) plant; (**B**) flower; (**C**) ripe fruits; (**D**) seeds.

**Figure 2 molecules-27-04161-f002:**
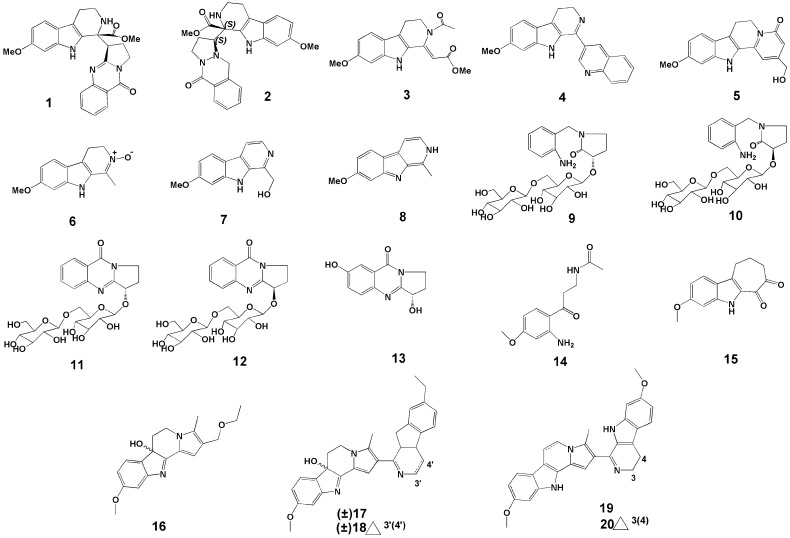
Structures of the twenty new compounds from *P. harmala*.

**Table 1 molecules-27-04161-t001:** The antimicrobial mechanism of *P. harmala*.

Species	Name	Composition	Mechanism	Reference
Bacteria	*S. aureus*, *P. aeruginosa*, *K. pneumoniae*	PhAMP	Disrupt the developed biofilm	[33]
	*P. aeruginosa*	PhAMP	Downregulate the expression of *flgK*, *pilA*, *cupA1*, *plsA* genes	[56]
	*S. aureus*	PhAMP	Upregulate the expression of *CPS5* and *icaA*	[56]
	*R. solanacearum*	*P. harmala* seeds	Cause cellular damage, clotting of the genome, as well as disorganized cytoplasm, and thickened cell wall	[37]
	*E. coli*	harmaline derivatives	Cause significant generation of ROS	[64]
	*Acinetobacter baumannii*	whole plant	Inhibit both the biofilm formation and the production of violacein pigment	[65]
Fungi	*B. cinerea*	harmol	Cause membrane integrity loss, cell wall disruption, and cytoplasm disorganization	[45]
	*A. flavus*	whole plant	Downregulate *aflR*, *aflM* and *aflP* genes	[66]
Virus	Bohv-1	harmine	Inhibit BoHV-1 replication	[67]
	HIV-1	*P. harmala* seeds	Inhibit reverse transcriptase	[48]
	HSV-2	harmine	Downregulate cellular NF-κβ and MAPK pathways	[68]

## Data Availability

Not applicable.

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
