# Peer review of "Antibacterial, Antifungal, Antiviral, and Antiparasitic Activities of Peganum harmala and Its Ingredients: A Review"

_molecules, 2022, doi:10.3390/molecules27134161_

Round 1
Reviewer 1 Report
It is very well-marked that this study is acceptable with minor revision and useful for publish in this journal.

Author Response
Point 1: Question about the writing of P. harmala (line 15), Peganum harmala Linn (line 50), T. annulate (line 396) and T. annulate (line 399)
Response 1: Thanks to the expert for your careful review. The writing had been correctly corrected in the paper. We modified P. harmala (Line 15) to Peganum harmala Linn, Peganum harmala Linn (line 50) to Peganum harmala L., T. annulata (line 396) to Theileria annulata, T. annulate (Line 399) to T. annulata.
Point 2: Suggestion about the replacing the article “Determination of therapeutic potential of Mentha longifolia ssp.Longifolia” with references in the article [1,2] (line 36) and “Phenolic content and biological activities of Rhus coriaria var. zebaria” with references [6,7] (line 43)
Response 2: The first reference you recommended is mainly to study the antioxidant activity, antimicrobial activity and oxidative stress properties of ethanolic extract of Mentha longifolia, and the second mainly studies the biological activity and phenolic content of Rhus coriaria L. var. in Duhok region in Iraq. They are about natural products of plants with special biological activities. We think they are more appropriate to be cited on lines 47 and 49. So we cited them there.
Point 3: The necessity of “Web of Science, PubMed, Google Scholar, Elsevier, and Chinese National Knowledge Infrastructure (CNKI)” (line 66, line67)
Response 3: The databases of Web of Science, PubMed, Google Scholar, Elsevier, and Chinese National Knowledge Infrastructure (CNKI) listed in manuscript are necessary. Because it indicated that all the information in the present manuscript was come from these databases.

Reviewer 2 Report
After checking the paper entitled "Antibacterial, antifungal, antiviral and antiparasitic activities of Peganum harmala and its ingredients: a review", my opinion is that this paper is a scientific-relevant and represents the literature for understanding basic as well as advanced data of the targeted topic.
My opinion is that this paper can pass only a minor revision, so my suggestions are listed here:
- check the Reference list and coordinate with the Instruction for the Authors
- line 185-186 check names of microorganisms Cereus, Coli, Aureus need to be written with the small initial letter
- In Section 2, indicate what are bold numbers for a better understanding of the connection with Figure
- Expand the Figure 1 on a whole page
Author Response
After checking the paper entitled "Antibacterial, antifungal, antiviral and antiparasitic activities of Peganum harmala and its ingredients: a review", my opinion is that this paper is a scientific-relevant and represents the literature for understanding basic as well as advanced data of the targeted topic.
My opinion is that this paper can pass only a minor revision, so my suggestions are listed here:
Response: Thank you very much for your kindly comments
- check the Reference list and coordinate with the Instruction for the Authors
Response: Thank a lot. References have been carefully checked three times to ensure coherence.
- line 185-186 check names of microorganisms Cereus, Coli, Aureus need to be written with the small initial letter
Response: Thank a lot. The words “Cereus, Coli, Aureus” (line 185-186) had been written with small initial letter in the paper.
- In Section 2, indicate what are bold numbers for a better understanding of the connection with Figure
Response: Thank a lot. We have marked the bold numbers in Section 2, for a better understanding of the connection with Figure 2.
- Expand the Figure 1 on a whole page
Response: We have enlarged Figure 1 (Figure 2 in revised manuscript) and rearranged the order of the structural formulas. The structure of each compound can be seen more clearly.

Reviewer 3 Report
1. The review paper "Antibacterial, antifungal, antiviral and antiparasitic activities of 2 Peganum harmala and its ingredients: a review" dealt in detail with the subject matter, as evidenced by the number of 47 articles that examined antibacterial, antifungal, antiviral and antiparasitic properties. In addition, the authors also briefly discussed the effect of gene expression, where information was taken from 4 articles, and one article on reverse transcriptase.
2. The paper is adequately prepared and the topics covered are current research topics in many academic centres. The amount of other work interpreted is sufficient for the paper to be a review.
3. Review papers done nowadays should be more eye-friendly, they cannot rely only on the text and its interpretation, especially in such a reputable journal as Molecules. I urge authors to include colour photographs of plants and their parts, these elements will make the work more attractive and enjoyable to look at
Author Response
Point 1: The review paper "Antibacterial, antifungal, antiviral and antiparasitic activities of Peganum harmala and its ingredients: a review" dealt in detail with the subject matter, as evidenced by the number of 47 articles that examined antibacterial, antifungal, antiviral and antiparasitic properties. In addition, the authors also briefly discussed the effect of gene expression, where information was taken from 4 articles, and one article on reverse transcriptase.
Response 1: Thank a lots.
Point 2: The paper is adequately prepared and the topics covered are current research topics in many academic centres. The amount of other work interpreted is sufficient for the paper to be a review.
Response 2: Thank a lots.
Point 3: Review papers done nowadays should be more eye-friendly, they cannot rely only on the text and its interpretation, especially in such a reputable journal as Molecules. I urge authors to include colour photographs of plants and their parts, these elements will make the work more attractive and enjoyable to look at.
Response 3: Thank you for your kindly suggestion. We inserted a section about biological characteristics of P. harmala after introduction and four photographs about Peganim harmala, flower, fruits and seeds were attached (see Figure 1 in revised manuscript).
